**Cite this article:** van Delft MAM, Cegiel A, van Egmond M and Boon L (2025). Biomarker understanding first lessons from drug development for IgA-driven autoimmune and fibrotic diseases. *Cambridge Prisms: Precision Medicine*, **3**, e8, 1–17

personalized medicine; biomarker; IgA; CD89; autoantibodies

**Corresponding author:**
Louis Boon;
Email: louis.boon@jjp.bio

# Biomarker understanding first lessons from drug development for IgA-driven autoimmune and fibrotic diseases

Myrthe Alexandra Maria van Delft[1,2,3,4], Aleksandra Cegiel[1],
Marjolein van Egmond[3,4,5] and Louis Boon[1,2]

[1]JJP Biologics, Warsaw, Poland; [2]Abyra BV, Leiden, Netherlands; [3]Amsterdam UMC location Vrije Universiteit Amsterdam, Molecular Cell Biology and Immunology, De Boelelaan 1117, Amsterdam, the Netherlands; [4]Amsterdam institute for Immunology and Infectious diseases, Immunology, Amsterdam, the Netherlands and [5]Amsterdam UMC location Vrije Universiteit Amsterdam, Surgery, De Boelelaan 1117, Amsterdam, the Netherlands

## Abstract

The concept of personalized medicine and its significant benefits for patients and society was introduced over three decades ago. The Human Genome Project (initiated in 1990 and completed in 2003) greatly accelerated the development of precision medicine. In many cancers, defined biomarkers are used to select patients for therapy. For example, KRAS mutations are used to guide treatment with Sotorasib, while tumor expression of (wild type) human epidermal growth factor receptor 2 and 3 (HER2 and HER3) are used to select patients for trastuzumab and cetuximab, respectively. Nonetheless, the clinical adoption of companion diagnostics to facilitate a patient-centric approach in inflammatory diseases remains disappointing. One key reason why the development of companion diagnostics may be delayed autoimmune and fibrotic diseases can be the timing when clinical development teams inform R&D teams about relevant biomarkers or companion diagnostic to select patients, disease monitoring or treatment termination decisions. For clinical practicality, it is highly preferred to measure a biomarker in the systemic circulation, as blood samples can be obtained relatively easily in most diseases. However, discovering systemic biomarkers during clinical development has proven extremely challenging. Here, we describe an alternative approach, which we have used to select the most appropriate target for IgA driven autoimmune and fibrotic diseases. In this specific context, autoantigen-specific assays to determine autoantibody serum levels are widely available for a variety of indications. A detailed analysis of the biological pathways that affect the biomarker can uncover multiple potential therapeutic targets, allowing selection of the most optimal target from a clinical development perspective. Identification of a relevant biomarker before clinical development is initiated, enabling patient stratification in early clinical studies. Selection of the appropriate patient population based on biomarker presence reduces the number of patients needed and consequently, clinical development costs. Moreover, such a patient stratification approach minimizes the risk of including patients who are unlikely to respond, thereby avoiding unnecessary adverse events. This approach was applied during the selection of an anti-CD89 antagonist monoclonal antibody for IgA-mediated autoimmune and fibrotic diseases, serving as an illustrative example of this novel strategy.

## Impact statement

Precision medicine based on biomarker presence is important for selecting the appropriate patient population. Before using a biomarker, biomarker understanding is important. This will guide us which therapeutic target is optimal to reduce pathology and helps us in the development of a drug specific for this target. Developing a drug accompanied by a relevant biomarker to stratify patients, results in a precision medicine guided patient-centric approach.

## Literature retrieval strategy

First, a clear topic, scope and outline for the review were defined. Several diseases were selected for an extensive literature search. For this search, MEDLINE and PubMed were used, applying inclusion criteria such as mice models, CD89 or FcαR(I), disease name, IgA, fibrosis, autoimmune disease and autoantibodies. No restrictions were placed on publication dates, as the field of IgA/ FcαRI research is relatively small and the number of publications is limited.





## Introduction

Digital navigation has become an integral part of our daily life, guiding us effortlessly to our destinations. Whereas the sun, the moon and compass once offered general directions for travelers, these are by no means comparable to the level of personalized guidance provided by current Global Positioning System (GPS) navigation systems. Similarly, navigation for inflammatory patients through the health care system and treatment options has been limited by population-based medicine. Like the compass, this approach offers general and no personalized guidance, assuming that patients with a particular condition share the same disease. Consequently, drug development historically focused on a 'one-treatment-fits-all' solution for every disease. This approach has successfully delivered multiple effective medicines, changing the life expectancy and quality of life of many patients but falls short for patients with therapy-resistant or atypical disease forms. We now realize the urgent need to move beyond this symptom-based 'one-treatment-fits-all' classification of patients towards a more patient-centric approach in which biomarker information is used to identify patients that are likely to respond to the treatment (Valla et al., 2021). In this sense, the landscape of medical treatments is undergoing a transformative change. Despite the potential of precision medicine in healthcare (Schilsky, 2010), by individualizing diagnostics and treatments according to each patient's uniquely evolving disease status, it is not incorporated in our healthcare system. Moreover, the use of companion diagnostics to facilitate a patient-centric approach remains disappointing for inflammatory diseases (Oliner et al., 2025), especially in the context of drug development. This is mainly caused by the timing that clinical development teams start informing R&D teams about the existence of a biomarker/companion diagnostic to stratify patients, to follow their disease course and eventually to terminate their treatment, namely late in clinical development. For ease of clinical use, it is highly preferred to measure such a biomarker in the systemic circulation, as a blood sample can be obtained relatively easily in most diseases. However, to discover such a systemic biomarker during clinical development is extremely difficult. We here describe an alternative approach to identify and understand the biology of the biomarker before clinical development is initiated, so patient stratification can be applied in early stages of clinical studies. Selection of the appropriate patient population using such a biomarker would reduce the number of patients needed in clinical studies and thus reduce the costs of clinical development. Moreover, patient stratification prevents adverse events in patients that were, based on such a biomarker analysis, unable to respond anyway. To facilitate such a strategy, a biomarker needs to be selected prior to the selection of the therapeutic target. Detailed analysis of the biological pathways that affect the biomarker will uncover several potential therapeutic targets from which, from a clinical development perspective, the most optimal target can be selected. The existence of a biomarker/companion diagnostic is of importance for providing the right drug, at the right dose, via the right route of administration, at the right time to the right patient. In this way, precision medicine holds the promise to deliver better medicines, improve patient outcomes and lower healthcare costs. It has the potential to benefit millions of patients and save annually billions of dollars through new and better targeted therapeutic options.

## Biological pathway and biomarker selection; targeting the IgA-CD89 axis

As an example of biomarker and pathway selection, we focus on the development of an anti-CD89 antagonistic antibody to target the IgA-CD89 axis in a subgroup of treatment-refractory patients in various autoimmune and fibrotic indications (van Delft et al., 2023). The IgA pathway generated our interest because of clinical observations that patients with high IgA autoantibody levels suffer from more severe disease in various autoimmune and fibrotic indications and are over-represented in the treatment-refractory population (Bobbio-Pallavicini et al., 2007a; Sakthiswary et al., 2014; Sowa et al., 2018; Solomon et al., 2020; Yadav et al., 2020; Sieghart et al., 2022). Furthermore, the IgA Fc receptor (CD89/FcαRI) is expressed on cells of the myeloid lineage, including neutrophils, monocytes and eosinophils, which are abundantly present in affected and inflamed tissues (reviewed in (Aleyd et al., 2015; Cecchi et al., 2018; Yap et al., 2018; Breedveld and van Egmond, 2019; van Gool and van Egmond, 2020). We previously showed that IgA immune complexes are potent stimuli to activate myeloid cells (van der Steen et al., 2009; Aleyd et al., 2014; Breedveld et al., 2021; van Delft et al., 2023), which may drive tissue damage and chronic inflammation. The potent activation of these cells via the IgA-CD89 axis reinforced the decision to therapeutically target this pathway. Another key factor in selecting the IgA-CD89 axis was the practical feasibility of developing a companion diagnostic. Autoantigens have been identified in many autoimmune and fibrotic diseases, and presence of autoantibodies are routinely determined in diagnostic or hospital laboratories. Consequently, a potential companion diagnostic is already established and available. However, in most clinical settings autoantibody assays have only been developed to measure IgM or IgG autoantibodies, despite the correlation between IgA autoantibodies and more severe disease (Bobbio-Pallavicini et al., 2007a, 2007b; Prüss et al., 2012; Doss et al., 2014; Sakthiswary et al., 2014; Ten Klooster et al., 2015; Jendrek et al., 2017; Yadav et al., 2020; Wunsch et al., 2021; Sieghart et al., 2022; Arai et al., 2025). Developing IgA-specific assays, based on already established IgG or IgM platforms would be relatively straightforward, supporting the view that the IgA-CD89 pathway has the potential to develop companion diagnostics for each of the disease indications to stratify patients eligible for inhibition of the IgA-CD89 axis. Moreover, measuring the amount of antigen-specific dimeric IgA in serum could be of importance to determine the active state of the disease (van Mourik et al., 2025). To establish an IgA matrix on the affected tissue, dimeric IgA has, due to its higher avidity than monomeric IgA and IgG, a competitive advantage. Next to the presence of the IgA also the relative contribution of the IgA autoantibodies to the total autoantibody response will be important, because for proper immune activation through CD89, crosslinking of CD89 is crucial. For this reason, the density of IgA autoantibodies on the tissue will determine if CD89-mediated activation is driving the disease. Data indicated that a ratio of 1:4 of IgA:IgG in complex form is still sufficient to induce NETosis by human neutrophils, which could be completely inhibited by anti-CD89 in vitro (Gimpel et al., 2021). Thus, patient selection in clinical trials may be based on autoantibody titer or other IgA-related characteristics, such as the IgA:IgG ratio, degree of IgA dimerization, or glycosylation profile.

## IgA; Structure, location and functional activity

During immune responses, B cells undergo isotype switching, where they change the class of antibody they produce while retaining antigen specificity. This typically progresses from IgM to downstream isotypes, such as IgG, IgE, or IgA, depending on the cytokine signals present in the immune environment. Notably, IgA, particularly IgA2, is the final isotype produced in this sequence, often reflecting a more mature or chronic stage of immune

responses (Stavnezer and Schrader, 2014). Humans express two closely related IgA subclasses, IgA1 and IgA2, whereas most other mammals produce only a single IgA subclass that resembles IgA2 (de Sousa-Pereira and Woof, 2019). IgA1 and IgA2 differ in their hinge region and number of glycosylation sites (Woof and Russell, 2011). IgA1 contains three to six O-linked glycans in its hinge region, while IgA2 is devoid of O-linked glycosylation. Alterations in O-linked glycosylation of IgA1 can cause conformational changes resulting in increased pathogenic immune complex formation, which is a key factor in diseases like IgA nephropathy (Novak et al., 2018). Next to the different subclasses, in humans, IgA exist in several structural forms; i.e. monomeric IgA, dimeric IgA, polymeric (including pentameric) IgA and secretory IgA (Woof and Russell, 2011; de Sousa-Pereira and Woof, 2019). Serum IgA is mostly monomeric and produced by plasma cells in the bone marrow, spleen, and lymph nodes. It consists predominantly of the IgA1 subclass with an IgA1:IgA2 ratio of 9:1 (Woof and Russell, 2011; de Sousa-Pereira and Woof, 2019). In contrast, IgA at mucosal sites is predominantly dimeric and produced by local plasma cells in the lamina propria. Dimeric IgA consists of two monomers that are linked tail-to-tail with a joining (J-) chain. Dimeric IgA is transported through epithelial cells by the polymeric Ig receptor (pIgR) and secreted at the luminal site (Horton and Vidarsson, 2013). During this process, part of the pIgR, referred to as secretory component, remains attached to dimeric IgA forming secretory IgA. In mucosal tissues, IgA1 and IgA2 are more evenly distributed.

The human immune system prioritizes production of IgA, since more IgA is produced daily compared to all the other classes of immunoglobulins combined. At 2–3 g/L IgA is the second most prevalent circulating Ig in serum after IgG. Traditionally, monomeric IgA has been described as involved in systemic, non-mucosal immune responses. In contrast, dimeric IgA is classically described as mucosal antibody, which is locally produced at the lamina propria. This is, however, an oversimplification, as antigen specific dimeric IgA has been detected in serum after non-mucosal and mucosal immunizations (Eijgenraam et al., 2008). Furthermore rheumatoid factor (RF)-IgA is detected as dimeric IgA in serum, synovial fluid and saliva of patients with rheumatoid arthritis (RA) (Otten et al., 1991). However, at mucosal surfaces and in external secretions IgA is the dominant isotype and serves as the first line of defense against invading pathogens (23). The mucosal immune system—particularly in the gastrointestinal and respiratory tracts—maintains a high level of immune activation due to constant exposure to pathogens and the commensal microbiome. This leads to the continuous production of pathogen and microbiome specific dimeric IgA by antigen specific B cells. In contrast, breast tissue, which lacks exposure to a microbiome and does not maintain a locally stimulated immune environment, still secretes dimeric IgA into breast milk (reviewed in [Brandtzaeg, 2010; Hurley and Theil, 2011]). This dimeric IgA is likely derived from immune responses initiated at distant sites, not locally within the breast tissue, as IgA autoantibodies and vaccine mediated IgA antibodies can be detected in breast milk (Zingone et al., 2017; Zhao et al., 2023). From an evolutionary perspective, the secretion of dimeric IgA in breast milk is crucial for neonatal immunity, providing passive protection by neutralizing pathogens in the gastrointestinal tract of the newborn. However, what the effect is of autoantibody IgA to the neonatal immunity is unknown and poorly studied. Overall, these observations suggest that dimeric IgA reflects an active immune response to a specific antigen, rather than exclusively indicating a mucosal origin. For example, dimeric IgA antibodies specific to mucosal pathogens like *E. coli* can be detected in serum2020; 2018 (32, 33).

As mentioned above, IgA is the most abundantly produced antibody class of the human body (Pabst and Slack, 2020) and we previously demonstrated that it is extremely potent in activating myeloid cells (like neutrophils, monocytes, eosinophils and osteoclasts) (van der Steen et al., 2009). Neutrophils are by far the most abundant immune cells in the human circulation, comprising approximately 60–70% of the total white blood cell count (Tigner et al., 2025). Although their presence in high numbers and their potent effector functions, surprisingly little therapeutic interventions are directed towards neutrophils. IgA potently activates neutrophils via cross-linking of CD89 through IgA immune complexes. This results in the fast release of leukotriene B4 (LTB4), a potent chemoattractant that promotes rapid recruitment of more neutrophils from the circulation into affected tissues (van der Steen et al., 2009). Neutrophils are also a source of various chemokines, which play a critical role in recruitment of other leukocyte subtypes, thereby amplifying local inflammation, e.g. during skin inflammation (Su and Richmond, 2015). Excessive neutrophil activation results in the formation of neutrophil extracellular traps (NETs). While NETs have an antimicrobial defense mechanism, their unwarranted formation can dysregulate innate and adaptive immune functions, including increased production of inflammatory cytokines and loss of tolerance to self-antigens (Gao et al., 2025). During NETs formation, histones and other intracellular antigens are released into the extracellular matrix, and can be captured by dendritic cells, which process and present them as antigens (Sangaletti et al., 2012), promoting synthesis of the next generations of autoantibodies (Pruchniak et al., 2015; Lee et al., 2017). Thus, this self-perpetuating cascade may enhance further autoreactive IgA antibody production, neutrophil activation, NETs formation, and exacerbation of autoimmune responses, including tissue inflammatory cell infiltration. In addition, neutrophil-derived proteases, released during NETs formation, can process pro-cytokines into mature cytokines and cleave cytokine receptors (e.g. IL-6 receptor) enabling trans-signaling in neighboring cells, further exacerbating disease (Thieblemont et al., 2016).

We showed that blocking the interaction between IgA-autoantibodies and CD89 with an antagonizing anti-CD89 antibody inhibited IgA-mediated activation of neutrophils and eosinophils, thereby preventing the induction of NETosis, phagocytosis and production of cytokines and chemokines (van der Steen et al., 2012; van Delft et al., 2023). Consequently, recruitment of cells to inflammatory sites is reduced, halting tissue damage and breaking the vicious cycle of IgA-driven inflammation.

## In vivo mouse disease models for the IgA-CD89 axis

Species differences in IgA and CD89 biology significantly present a major limitation for developing relevant in vivo models to study IgA-driven human pathologies. CD89 has been identified in several monkey species, horses, cattle, hamsters, gerbils, and rats (Maruoka et al., 2004; Morton et al., 2005). However, mice, which are mostly used to study immunological processes, CD89 is absent due to a gene translocation (Reljic, 2006). Consequently, murine models using wild type mice are not suitable to study the functional consequences of IgA autoantibodies in the context of human disease. Any findings from such models can be considered as non-representative, misleading, or non-translatable to human immunopathology. Moreover, mouse data showing potential immune modulatory effect of systemic IgA should be interpreted with caution, because in the absence of CD89, IgA does not have the

same pathological consequences as in humans, as IgA in mice cannot mediate pro-inflammatory effects in the absence of CD89 receptor. The dampening effect of mouse IgA can occur because mouse IgA autoantibodies, in the absence of their receptor, compete for autoantigens with IgG and IgM for which the receptors are present in mice. Consequently, despite the well-established clinical importance of IgA in diseases such as IgA vasculitis (IgAV) and IgA nephropathy (IgAN) where IgA is explicitly named in the disease terminology—the pathogenic role of IgA autoantibodies remains understudied and undervalued in preclinical models of human disease. The use of wild type mice, which lack CD89, to study the pathogenicity or functional consequence of systemic IgA-complexes in e.g. IgAV or IgAN disease models is therefore difficult to justify.

To investigate the functional consequences of these IgA-complexes three different human CD89-transgenic (CD89-Tg) mice models have been developed. First, we generated a cosmic clone containing the CD89 gene and human regulatory elements was used, resulting in expression of human CD89 protein on neutrophils, a sub-population of monocytes and on activated macrophages (van Egmond et al., 1999). Secondly, CD11b (Launay et al., 2000) or thirdly, CD14 (Xu et al., 2016) driven expression of CD89 both resulted in expression of human CD89 on monocytes and macrophages. The pathogenic role of soluble CD89–IgA complexes in IgAN was demonstrated in the CD11b driven model by transferring disease to RAG-2$^{-/-}$ recipient mice via serum from CD89-Tg donors. Disease was not transferred when serum from control non transgenic mice was used, or when serum from CD89-Tg mice had been adsorbed with anti-CD89 mAbs (Launay et al., 2000).

Our studies were further complicated due to the inability of mouse IgA to bind to human CD89, human CD89-Tg mice therefore only permit to study the interaction between exogenously administered human IgA and CD89-expressing cells, which is hampered by the short half-life of exogenous administrated human IgA (Heemskerk et al., 2021; Bos et al., 2022; van Delft et al., 2023). Moreover, since murine IgA does not bind human CD89, this restricts the physiological relevance of endogenous IgA responses in these models (van der Boog et al., 2004). Only when endogenous disease-specific IgA autoantibodies are induced that can bind to human CD89 in these transgenic models, functional and translation results can be obtained in mouse models. The absence of appropriate disease models and consequently, misleading conclusions from non-relevant in vivo experiments studying IgA biology in the absence of CD89, may explain the limited therapeutic interest to target IgA or CD89. This is particularly surprising, given the potential pathogenic involvement of the IgA-CD89 axis in a subgroup of patients in many autoimmune and fibrotic diseases (reviewed in [Breedveld and van Egmond, 2019]), which emphasizes the importance of identifying these patients for stratification and the need of a precision medicine approach using patient samples.

## Options to interfere in the IgA-CD89 axis

To identify a suitable target for intervention in the IgA-CD89 pathway, various strategies were considered. An approach that was considered, but immediately rejected, was to interfere with the pathogenic activity of IgA autoantibodies by blocking the autoantibody response in an antigen-specific manner. In theory, antigen-specific targeting would be optimal when an autoimmune response is limited to a single antigen and epitope, as in this way not only IgA, but also IgG and IgM autoantibody responses are inhibited. However, this strategy is impractical, since in most autoimmune diseases diverse autoantigens have been reported. Even within a single disease indication multiple antigens or epitopes have been described to contribute to the pathogenicity of the disease. For instance, in rheumatoid arthritis (RA), rheumatoid factor (RF), anti-citrullinated protein antibodies (ACPA), anti-double stranded DNA (anti-dsDNA) antibodies and anti-carbamylated protein (anti-CarP) antibodies have been found (Yadav et al., 2020; Steiner and Toes, 2024). Furthermore, even with a single autoantigen different epitopes have been identified (Oskam et al., 2023). This complexity undermines the feasibility of an antigen-dependent blocking strategy in the IgA–CD89 axis, leading to the rejection of this option. Another approach that was considered was to block IgA at the site where it binds to CD89, thereby physically preventing the IgA-CD89 interaction. This concept is in analogy with the mechanism of action of omalizumab (Xolair), a humanized monoclonal antibody which binds to the Cε3 domain of immunoglobulin E (IgE) (Wright et al., 2015). Omalizumab forms complexes with IgE preventing binding to FcεRI, thereby averting IgE-mediated allergic inflammation. Similarly, for IgA this strategy would involve targeting the constant domain of IgA at the position where it binds to CD89 and thus preventing binding of IgA and IgA complexes to CD89 (Breedveld and van Egmond, 2019). Although this approach works very well in the case of IgE, an isotype of which only a limited amount is present in the systemic circulation, the high levels of IgA in the systemic circulation and in the mucosa might be challenging. IgA is the most abundant immunoglobulin in the human body and is produced at approximately 60 mg/kg/day. Furthermore, although omalizumab prevented IgE-driven allergic and asthmatic responses by approximately decreasing 90% of free IgE, the total level of anti-IgE-IgE complexes significantly increased and remained elevated for up to one year after discontinuation of treatment (Busse et al., 2001). Since IgA immune complexes may be elevated in many autoimmune diseases, targeting IgA may further increase the size of IgA-containing immune complexes. For example, IgA-nephropathy (IgAN) is the most common glomerulonephritis worldwide, which is characterized by the deposition of IgA containing complexes in the glomerular mesangium of the kidneys. Similar pathogenic IgA deposition is observed in systemic lupus erythematosus (SLE) with lupus nephritis, in IgA vasculitis (IgAV) affecting small vessels, and in rheumatoid arthritis (RA), where IgAN may arise as a comorbidity alongside elevated rheumatoid factor (RF) levels (Makino et al., 2002; Azegami et al., 2025). Given these concerns, particularly the risk of increasing pathogenic IgA immune complexes, which in theory might induce IgAN or IgAV in asymptomatic individuals, this treatment strategy was also rejected. Therefore, a third strategy, i.e. targeting CD89, was selected and we initiated the development of an antagonistic anti-CD89 antibody to inhibit pathogenic IgA responses in various autoimmune and fibrotic diseases. For this, we have immunized mice with human CD89 and the obtained mAbs were screened for the ability to block IgA binding to a CD89-expressing cell-line. Various mAbs were selected and although these anti-CD89 mAbs bind to different epitopes on EC1 of CD89, they all have the capacity to inhibit IgA-mediated phagocytosis, NET release and neutrophil migration. Moreover, IgA mediated LTB4 and lactoferrin release are decreased in supernatant from anti-CD89 mAbs-treated neutrophils. More detailed studies we have performed revealed that the epitope on EC1 on which the lead mAbs bind was crucial to prevent either

neutrophilic apoptosis or unexpected cross-reactivity in other human tissues 2023. An additional advantage of anti-CD89 therapy, by which IgA autoantibodies become non-pathogenic, is that these IgA autoantibodies occupy antigen-specific epitopes in the tissue. These IgA autoantibody in the absence of CD89 effector functions compete with IgG and IgM antibodies for binding in tissue, thereby also limiting the pathogenic potential of IgG and IgM.

## IgA-mediated disease indications

Based on clinical observations that patients with high IgA auto-antibody levels experience more severe disease in various auto-immune and fibrotic indications, inhibition of the CD89-IgA axis may be justified in these indications (Bobbio-Pallavicini et al., 2007a; Sakthiswary et al., 2014; Sowa et al., 2018; Solomon et al., 2020; Yadav et al., 2020; Sieghart et al., 2022). The use of an anti-CD89 antagonist for a given indication depends on the availability of currently approved drugs and molecules in development for that specific condition. These drug/molecules are listed per indication in Table 1.

### *Linear IgA dermatosis (LAD); a rare disease indication incorporating the companion diagnostic*

Linear IgA Dermatosis (LAD) is an ultra-rare autoimmune blistering skin disorder characterized by the presence of IgA autoantibodies against antigens of the skin's basement membrane zone (Cozzani et al., 2020). Histopathological examination of LAD patients shows subepidermal bullae with CD89-expressing neutrophilic and/or eosinophilic infiltrates in the papillary dermis. The diagnosis of LAD can only be established by histopathological and immunofluorescence findings. The differential immunofluorescence examination of perilesional skin shows linear IgA deposition. In this respect, dermatologists are ahead in the IgA-CD89 scientific field, as they identify LAD, representing a small subgroup of the total subepidermal bullous patient population, as a distinct entity with an IgA mediated mechanism of action. A mechanistic overview of LAD is presented in Figure 1.

The availability of an already established companion diagnostic in LAD. i.e. the detection of IgA autoantibodies in the skin, combined with the involvement of neutrophils and eosinophils (which both express CD89) as pathological effector cells, renders LAD as the perfect proof the concept indication to inhibit pathological IgA responses with our developed antagonizing anti-CD89 antibody. The ultrarare status of LAD may limit the number of available patients for clinical assessment. Nonetheless, due to the high specificity of patient selection, less patients will be needed to demonstrate therapeutic efficacy. Importantly, by choosing this ultrarare indication, for which no specific therapy is available, we demonstrate our commitment to precision medicine by addressing unmet medical needs in patient populations that are often overlooked in drug development—ensuring 'no patient is left behind'.

### *IgA vasculitis and IgA nephropathy; Systemic primary IgA-mediated diseases with no need for stratification*

The ultra-rare status of LAD helps explain the limited commercial interest in developing a specific approach to antagonize the devastating results of CD89-mediated neutrophil activation for this indication. It is however surprising that no efforts have been undertaken for IgAV and IgAN, which are two systemic indications in which IgA is central in both the nomenclature and pathophysiology. In both diseases, IgA containing immune complexes obstruct blood flow through smaller vessels or deposit in the kidney, respectively (Pillebout, 2021; Cambier et al., 2022). Like LAD, IgA deposits are part of the differential diagnosis of IgAV and IgAN and therefore there is no need for a companion diagnostic to stratify patients for treatment with anti-CD89 mAbs.

IgAV, also known as Henoch-Schönlein purpura, is the most common form of childhood vasculitis, but can also affects adults. Although the disease was already described over 200 years ago, its pathogenesis remains to be elucidated. The disease is characterized by IgA1-immune deposits, complement factors and neutrophil infiltration, which is accompanied by vascular inflammation (Heineke et al., 2017). In a subset of patients IgAV progresses into glomerulonephritis with symptoms comparable to IgAN that include hematuria, proteinuria and IgA deposition in the glomerulus, which ultimately can result in end-stage renal disease (Heineke et al., 2017).

IgAN is the most common primary glomerulonephritis that can progress to renal failure. In IgAN, immune complexes containing galactose-deficient (Gd-) IgA1 are found, which are thought to play a role in pathogenesis (Lai et al., 2016). IgAN is characterized by glomerular deposits of IgA immune complexes comprised of Gd-IgA1, complement C3, and variable amounts of IgG and/or IgM against Gd-IgA1 (Knoppova et al., 2016). Local immune reactivity to IgA immune complexes stimulates proliferation of mesangial cells, synthesis of extracellular matrix, and infiltration of inflammatory cells in glomeruli (Lai et al., 2016). The observations of recurrent IgA depositions in kidney grafts in transplant recipients with IgAN strongly suggest the extrarenal origin of pathogenic IgA immune complexes (Berthelot et al., 2015).

Previous studies have also focused on the role of soluble CD89 in IgA immune complexes and explored whether soluble CD89 levels in blood were associated with IgAN. Circulating IgA immune complexes are primarily cleared by the mononuclear phagocyte system through CD89 (Chen et al., 2018). It was suggested that, after binding to CD89, IgA complexes induce proteolytic shedding of CD89, resulting in circulatory soluble CD89-IgA complexes (Launay et al., 2000). Circulating immune complexes containing galactose-deficient IgA1, IgA1, IgG and/or soluble CD89 likely contribute to IgAN pathophysiology. These complexes travel from the circulation to the kidney where undergalactosylated polymeric IgA1 (and immune-complexes containing poorly galactosylated IgA1) interact with CD71 on mesangial cells (Moura et al., 2001; Haddad et al., 2003). This interaction increases CD71 expression, and induces transglutaminase (TGase)2, which can bind to sCD89–IgA1 complexes and directly interact with CD71. Ultimately this results in a self-amplifying loop of immune complex retention and mesangial inflammation (Daha and van Kooten, 2013).

The presence of sCD89 creates an interesting paradox. Shedding of CD89 from phagocytic cells inhibits effective IgA-complex clearing from the circulation, whereas incorporation of sCD89 into IgA-complexes will increase immune complex size and facilitate interactions with CD71 and TGase2. However, sCD89 will also interfere with binding immune complexes to CD89-expressing cells, thereby preventing activation. Thus, elimination of sCD89 from immune complexes will simultaneously reduce mesangial interaction and decrease immune complex size but also increase the risk of inducing major immune activation through CD89-expressing immune cells. We anticipate that an antagonistic anti-CD89 mAb will extract soluble CD89 from immune complexes due to its much higher affinity for CD89 than the affinity of IgA for CD89, thereby

**Table 1.** Approved drugs (EU/US) & phase 3 therapies

| Disease indication | Approved therapies – indicate territory | Targeted pathway | Mechanism / notes | Phase 3 / Late-stage pipeline |
|---|---|---|---|---|
| Linear IgA Dermatosis (LAD) | None specifically approved – EU/US. Off-label: dapsone; systemic corticosteroids; rituximab (for refractory cases). | Neutrophil activation; IgA-mediated injury | Dapsone inhibits neutrophil myeloperoxidase & chemotaxis; steroids broad immunosuppression; rituximab depletes CD20+ B cells. | No registrational (Phase 3) trials identified for LAD as of Oct 2, 2025. |
| IgA vasculitis (Henoch–Schönlein purpura) | None disease-specific – EU/US. Supportive care; off-label steroids ± immunosuppressants such as azathioprine / MMF / cyclophosphamide; rituximab (for refractory cases). | IgA immune-complex small-vessel vasculitis | Steroids anti-inflammatory; azathioprine/MMF/ cyclophosphamide reduce immune activation; rituximab depletes CD20+ B cells. | No disease-specific Phase 3 approvals; limited interventional studies—no large registrational trials reported as of Oct 2, 2025. |
| IgA nephropathy (IgAN) | Budesonide TR – EU (Nefecon®)/US (Tarpeyo®); Sparsentan (Filspari®) – US (accelerated); ACEi/ARB – US/EU. | Mucosal immune activation → pathogenic Gd-IgA1; proteinuria; renin–angiotensin–aldosterone system / Endothelin | Budesonide TR targets Peyer's patches to suppress IgA; sparsentan dual ETA/AT1 reduce proteinuria via hemodynamics.; ACEi/ARB decrease blood preseeure to protect the kidneys | Iptacopan (factor B inh.) APPLAUSE-IgAN – NCT04578834; Atacicept (TACI-Fc, APRIL/BAFF) ORIGIN 3 – NCT04716231; Sibeprenlimab (anti-APRIL) VISIONARY – NCT05248646; Zigakibart (anti-APRIL) BEYOND – NCT05852938. |
| Rheumatoid arthritis (RA) | csDMARDs (MTX, LEF, SSZ, HCQ) – US/EU; TNF inhibitors (adalimumab, certolizumab pegol, etanercept, golimumab, infliximab) – US/EU; Abatacept – US/EU; Rituximab – US/EU; IL–6R inhibitors (tocilizumab, sarilumab) – US/EU; Anakinra – US/EU; JAK inhibitors (tofacitinib, baricitinib, upadacitinib – US/EU; filgotinib – EU). | Immune activation; TNF-α; T-cell co-stimulation; B cells; IL-6; IL-1; JAK–STAT | As labeled: immune activation, cytokine blockade, T-cell co-stimulation inhibition, B-cell depletion, JAK inhibition. | No broadly novel mechanisms with positive Phase 3 beyond approved classes as of 2025; several BTK inhibitors discontinued/shifted focus. |
| Idiopathic pulmonary fibrosis (IPF) | Nintedanib (Ofev®) – US/EU; Pirfenidone (Esbriet®) – US/EU. | Profibrotic signaling (PDGFR/FGFR/VEGFR); TGF-β modulation | Nintedanib: multi-TKI; Pirfenidone: antifibrotic/anti-inflammatory with anti-TGF-β effects. | Nerandomilast / BI 1015550 (selective PDE4B inhibitor) FIBRONEER-IPF – NCT05321069; Prior Phase 3 failures: Pamrevlumab (anti-CTGF) ZEPHYRUS–1 – NCT03955146 (did not meet primary endpoint); Zinpentraxin alfa (PRM-151, PTX2) STARSCAPE – NCT04552899 (terminated early / no benefit). |
| Primary biliary cholangitis (PBC) | UDCA – US/EU; Obeticholic acid (Ocaliva®) – US/EU; Elafibranor (Iqirvo®) – US (accelerated, 2024) & EU (conditional, 2024); Seladelpar (Livdelzi®) – US (accelerated, 2024) / UK + EU 2025 updates vary. | Bile acid homeostasis; FXR agonism; PPAR δ/αδ | UDCA improves bile flow; OCA FXR agonist; Elafibranor dual PPAR-α/δ agonist, decreases the accumulation of bile acids in the liver; Seladelpar, PPAR-δ agonist decreasing the bile acid production in the liver. | Elafibranor ELATIVE – NCT04526665 (pivotal supporting approval); Seladelpar RESPONSE – NCT04620733 (pivotal). |
| MASH/NASH with fibrosis | Resmetirom (Rezdiffra™) – US (2024). | THR-β modulation (steatosis/inflammation/ fibrosis) | Resmetirom is a liver-directed THR-β agonist improving histology and lipids. | Efruxifermin (EFX, FGF21 analog) SYNCHRONY – e.g., NCT06528314; Semaglutide 2.4 mg ESSENCE – Phase 3 (design PMID 39412509); Tirzepatide Phase 3 programs progressing. |
| Cystic fibrosis (CF) | Ivacaftor; Orkambi®; Symdeko®/ Symkevi®; Trikafta®/Kaftrio® – US/EU (age & genotype dependent). | cystic fibrosis transmembrane conductance regulator (CFTR) chloride channel (potentiator + correctors) | Ivacaftor potentiator; lumacaftor/ tezacaftor/elexacaftor are correctors They all modulate the CFTR protein; triple therapy combines 2 correctors + potentiator. | Next-gen triple (vanzacaftor/ tezacaftor/deutivacaftor) – Phase 3 completed with non-inferiority vs. ETI (Keating et al., 2025) and pediatric expansion (Rubin, 2025). |
| COPD | Bronchodilators (LAMA/LABA) – US/EU; ICS/LABA – US/EU; Triple therapy – US/EU; Roflumilast (PDE-4) – US/EU; Dupilumab – EU (July 2024) and US (Sept 2024) for eosinophilic phenotype. | M3; β2-AR; PDE-4; airway/ type-2 Inflammation (IL-4/IL-13) | LAMA anti-M3; LABA β2 agonism; ICS anti-inflammatory; PDE-4 inhibition results in a decrease in inflammation; dupilumab blocks IL-4Rα to inhibit IL-4/IL-13. | Itepekimab (anti-IL-33) – mixed Phase 3 (AERIFY-1 met; AERIFY-2 missed) 2025; Tezepelumab (anti-TSLP) Phase 3 ongoing (e.g., NCT06883305); Tozorakimab (anti-IL-33) Phase 3 ongoing (AstraZeneca D9180C00003/00012). |

*(Continued)*

**Table 1.** (*Continued*)

| Disease indication | Approved therapies – indicate territory | Targeted pathway | Mechanism / notes | Phase 3 / Late-stage pipeline |
|---|---|---|---|---|
| Alzheimer disease (AD) | Symptomatic: donepezil/ rivastigmine/galantamine– US/EU; memantine – US/EU. Disease-modifying anti-amyloid mAbs: Lecanemab (Leqembi*) – US (traditional approval 2023; SC maintenance 2025); Donanemab (Kisunla™) – US (2024). | Cholinergic deficit; N-methyl-D-aspartate (NMDA) excitotoxicity; Amyloid-β | Acetylcholinesterase inhibitors prevent degradation of AcetylCholine; NMDA antagonist, decrease glutamate-induced brain cell damage; lecanemab/donanemab clear aggregated Aβ to slow decline. | Remternetug (LY3372993) – Phase 3 (e.g., NCT05463731; NCT06653153); ALZ-801 (valiltramiprosate) – APOLLOE4 Phase 3 NCT04770220 (APOE ε4/ε4). |
| Multiple sclerosis (MS) | Interferons, glatiramer (injectables) – US/EU; Orals: teriflunomide, fumarates, S1P modulators – US/EU; mAbs: natalizumab, ocrelizumab (incl. PPMS); Ofatumumab, Cladribine, alemtuzumab – US/EU. | Inflammation; Immune cell activation/ inflammation; Lymphocyte egress (S1P1); VLA-4; B-cell depletion and/or T cell depletion (CD20/ CD52; DHODH; NRF2) | Blocks trafficking, depletes B or T cells, reduces proliferation or immune reconstitution. | Fenebrutinib (BTK inhibitor) – Phase 3 in RMS (NCT04586023) and PPMS (NCT04544449) ongoing; Tolebrutinib (BTK inhibitor) – mixed Phase 3 (RMS negative; progressive MS positive – HERCULES). |

*Source*: GlobalData.

reducing immune complex size and their potential to interact with CD71 and TGase2. Moreover, an anti-CD89 antagonist will prevent activation of CD89-expressing cells by remaining IgA-containing immune complexes by blocking CD89.

### *Rheumatoid arthritis as an example of a systemic autoimmune disease indication for anti-CD89 therapy requiring stratification*

The presence of autoantibodies in the systemic (blood) circulation is a common feature of many inflammatory autoimmune disorders. Rheumatoid arthritis (RA) serves as a relevant example to illustrate the rationale and strategy to develop a companion diagnostic for systemic autoimmune disease in which a subset of patients has high levels of autoantigen-specific IgA autoantibodies. Although the pathogenic potential of autoantibodies (IgG, IgA and IgM), is widely recognized among rheumatologists, routine clinical diagnostics only measure autoantibody responses of the IgG and IgM class. With only minor adaptations these assays can be modified to also detect autoantibodies of the IgA class.

RA is a chronic autoimmune disease, affecting 0.5 to 1.0% of the global population. It is characterized by inflammation of synovial joints, leading to joint destruction and disability (Scott et al., 2010). The disease is hallmarked by the presence of autoantibodies such as RF, ACPA, anti-CarP antibodies and anti-dsDNA antibodies (van Delft et al., 2017; van Delft and Huizinga, 2020; Yadav et al., 2020). IgA-RF, IgA-ACPA and IgA anti-CarP are also detected in a dimeric form in sera and synovial fluid of RA-patients (Otten et al., 1991; van Delft, 2018, 2020), suggesting an active immune response. Interestingly, IgA-RF and IgA-ACPA are associated with the severity of RA, including joint erosion, disease progression and extra-articular manifestations (Berglin et al., 2006; Karimifar et al., 2014). The pathophysiological role of IgA autoantibodies in RA is supported by their interaction with neutrophils, which are abundantly present in synovial fluid of affected joints (Cecchi et al., 2018; Yap et al., 2018). We demonstrated that blocking the CD89-IgA interaction on neutrophils resulted in reduced NETs-release after stimulation with IgA-containing immune complexes from synovial fluid of RA patients (Aleyd et al., 2016). Moreover, stimulation of osteoclasts (bone-resorbing cells) with these complexes resulted in

IL-8 and IL-6 release. Additionally, stimulation of osteoclasts by IgA-complexes resulted in bone resorption, a pathogenic feature of RA (Breedveld et al., 2021). Anti-CD89 treatment inhibited IgA-mediated LTB4 production by neutrophils, which likely will terminate LTB4 mediated neutrophil recruitment to affected tissues (Figure 2) (van Delft et al., 2023). Currently, methotrexate is recommended as the first-line treatment of early RA. However, approximately 30% of patients do not respond to the treatment and require additional treatment, in most cases with TNF inhibitors (Bobbio-Pallavicini et al., 2007a; Potter et al., 2009; Sakthiswary et al., 2014; Sieghart et al., 2022). Blocking TNF has remarkably improved treatment outcomes in this subgroup of RA patients (Guo et al., 2018). Furthermore, anti-TNF agents are increasingly used in the management of an expanding number of rheumatic and systemic autoimmune diseases, such as, Crohn's disease, psoriasis, psoriatic arthritis, ankylosing spondylitis, Wegener granulomatosis and sarcoidosis. However, it was reported that RA-patients positive for IgA-RF or IgA-ACPA respond poorly to anti-TNF inhibitors (Bobbio-Pallavicini et al., 2007a, 2007b; Sakthiswary et al., 2014; Sieghart et al., 2022). Furthermore, IgA levels are increased in the therapy refractory subgroup of RA patients. Measuring IgA isotypes of RF and ACPA will provide prognostic information because their presence is associated with reduced responsiveness to TNF inhibitors. Only a subgroup of RA-patients exhibits high IgA autoantibody levels. Others have low titers of IgA autoantibodies or are negative, in which case the rationale for anti-CD89 is obsolete. Therefore, stratifying patients based on IgA autoantibody profiles is essential to determine who may benefit from anti-CD89 therapy. Also, patients having low auto-IgA titers which are negative for IgG might benefit of anti-CD89 therapy, as all antigen epitopes will be covered by IgA. In addition to identifying patients who are suitable for treatment, it is equally important to exclude patients who are unlikely to benefit from this therapy. During early clinical development this strategy enables more focused studies, involving only the relevant patients, resulting in smaller and more successful clinical studies. By minimizing the number of non-responders, fewer patients are required to achieve treatment effects. Furthermore, no unnecessary treatment is given to these low/no IgA positive patients, reducing potential side-effects and treatment costs. Moreover, measuring the IgA titer as companion diagnostic,

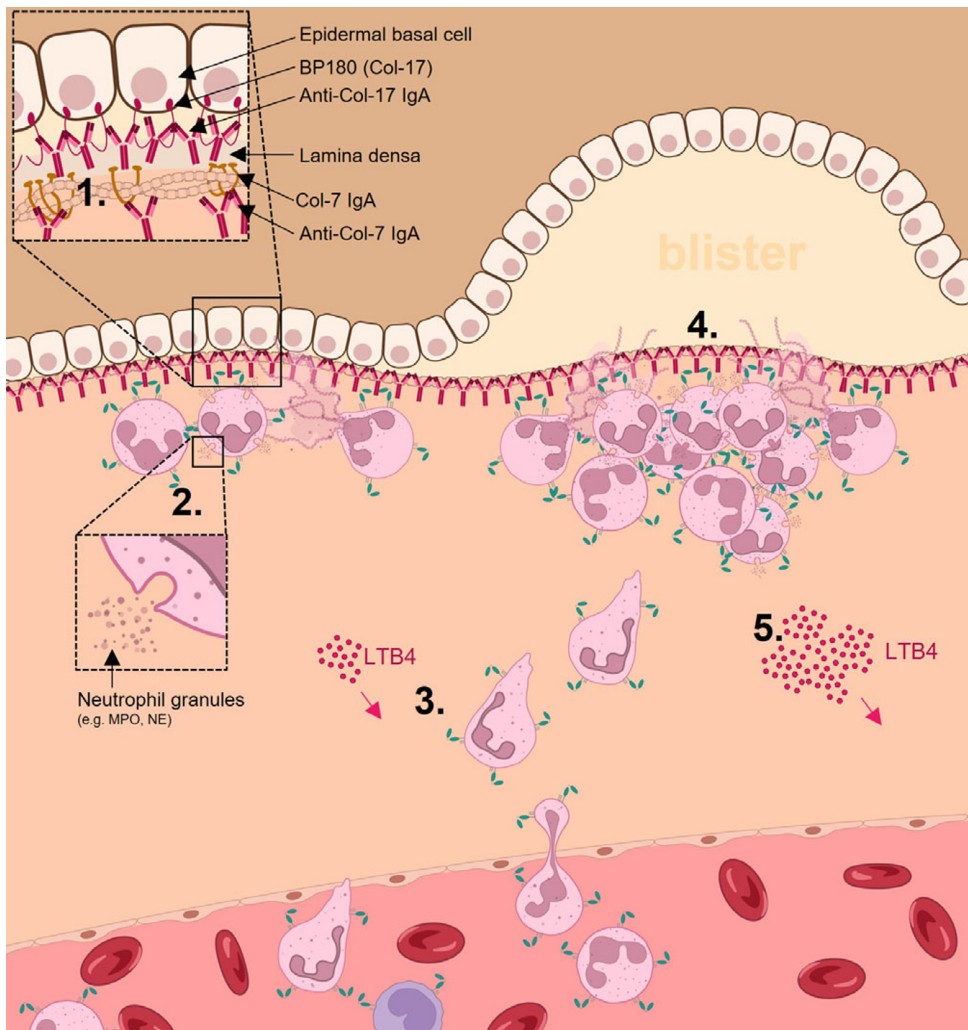

**Figure 1.** Schematic model of effects of IgA autoantibodies in inducing neutrophil (myeloid cell) activation and tissue damage in linear IgA dermatosis. IgA antibodies directed to Collagen type 17 (Col-17), or Collagen type 7 (Col-7) bind to the dermal/epidermal junction in the skin (1). These IgA-antigen complexes crosslink the FcαRI on neutrophils, resulting in cell activation and release of the potent neutrophil chemoattractant leukotriene B4 (LTB4) and interleuking-8 (IL-8) as well as myeloperoxidase and neutrophilic elastase (2). The LTB4 gradient leads to neutrophil attraction from the bloodstream and migration to site of inflammation (3). Neutrophil swarming and mass-activation leads to destruction of dermal-epidermal junction integrity and formation of blisters (4). Increased LTB4 production/gradient leads to the inflammatory amplification loop (5). Anti-CD89 binds to CD89 on neutrophils, displacing prebound IgA and preventing IgA mediated neutrophil activation (6). This will result in neutrophil apoptosis after their lifecycle, resolution of neutrophil swarming and subsequent restoration of dermal-epidermal junction integrity.

also provides physicians with a method to monitor treatment necessity over time. If IgA autoantibodies have been decreased in a patients' circulation, it may indicate that anti-CD89 therapy is no longer required, offering a rationale to discontinue treatment. Recently, it was reported that transmembrane (tm) TNF reverse signaling leads to transforming growth factor β (TGFβ) production in macrophages and that different anti-TNF agents trigger this pathway (Szondy and Pallai, 2017). Already in 1991, it was observed that TGFβ induced neutrophil recruitment to synovial fluid (Fava et al., 1991). Approximately 15 years later it was discovered that TGFβ induced class-switching towards IgA and IgA secretion (Stavnezer and Kang, 2009). This suggests that patients on anti-TNF therapy may be driven into IgA-mediated disease, supporting the measurement of IgA autoantibodies to enable stratification for treatment with anti-CD89 antagonistic antibody. The introduction of various biosimilar anti-TNF agents and the associated cost reduction resulted in earlier use of these therapies in the management of RA (Smolen et al., 2021). The DANBIO registry confirmed

that the sharp decrease in price of biosimilar TNF inhibitors has led to increased utilization (Jensen et al., 2020). The higher utilization of anti-TNF in RA or other inflammatory disease may inadvertently increase the prevalence of IgA-mediated disease via this TGFβ driven mechanism. Furthermore, high pretreatment levels of IgA RF are associated with a poor clinical response to TNF-inhibitors (Klaasen et al., 2011), there is sufficient justification to use IgA autoantibodies as companion diagnostic to stratify patients for treatment with antagonist anti-CD89.

Interestingly, tobacco smoking has been shown to aggravate disease in early RA, which is associated with increased seropositivity for IgA-RF (Papadopoulos et al., 2005; Manfredsdottir et al., 2006). Paradoxically, while smoking reduces the production of inflammatory cytokines (TNF, IFNγ and IL-6), patients have more severe disease, suggesting a different disease mechanism. The reduction of inflammatory cytokine levels may also explain why smokers do not respond very well to anti-TNF or anti-IL-6R mAbs (Baka et al., 2009). Moreover, smoking induces citrullination and

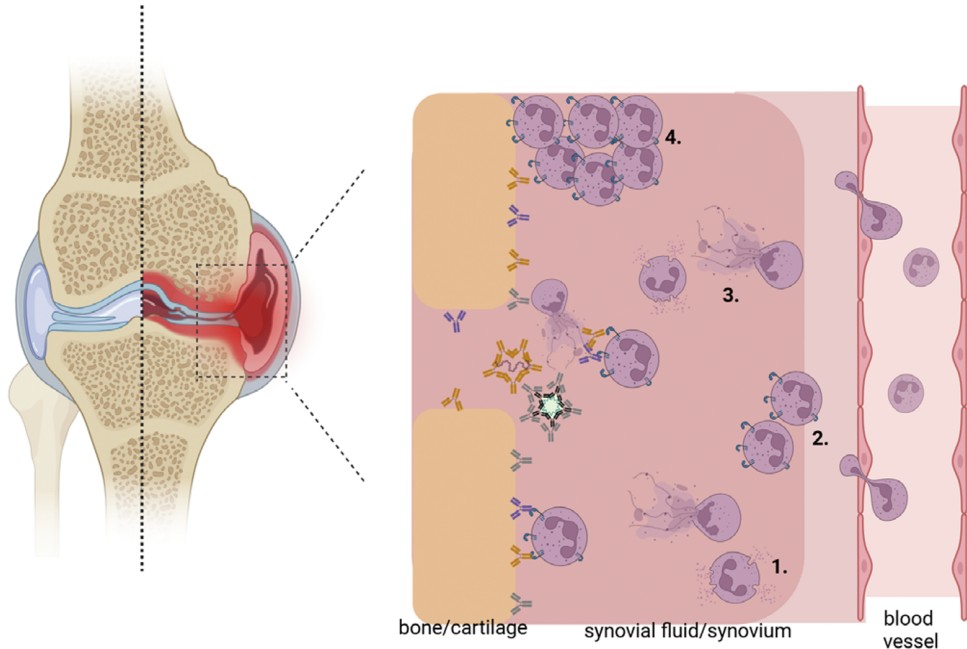

**Figure 2.** Schematic model of effects of IgA autoantibodies in inducing neutrophil (myeloid cell) activation and tissue damage in rheumatoid arthritis joints. IgA autoantibodies directed against citrullinated, carbamylated, human IgG Fc or dsDNA bind to carbamylated/citrullinated bone/cartilage, human IgG, DNA or NETs and form immune complexes. These IgA-antigen complexes crosslink the FcαRI on neutrophils resulting in cell activation, NETs release and release of the potent neutrophil chemoattractant leukotriene B4 (LTB4) and interleuking-8 (IL-8) (1). This way, a neutrophil migration loop is initiated (2), more neutrophils get activated (3) resulting in a massive influx of neutrophils in the synovium/synovial fluid (4), tissue damage and swollen joints in rheumatoid arthritis patients. Picture made by Biorender.

carbamylation (reviewed in [Kwon and Ju, 2021]) and thus ACPA and anti-CarP IgA. Hence, the reduction of smoking may have a preventive effect on the development of RF-IgA antibodies and consequently on disease activity. However, smoking may also be used as an initial stratification tool.

### Fibrotic lung diseases: Examples of immunologically driven mucosal disease indications for anti-CD89 therapy – "unknown cause or maybe not"

Since the activity and presence of antigen-specific IgA in immunologically driven mucosal diseases may differ from systemic IgA-mediated mechanisms, separate attention is given to lung diseases as examples for mucosally driven diseases. Like systemic auto-immune diseases, not all patients with distinct lung diseases have high IgA reactive autoantibodies. For this reason, the application of anti-CD89 antagonist therapy in these conditions also necessitates a stratified approach.

In medicine many pathologies are categorized as having an "unknown cause," often labeled as idiopathic to provide clinical terminology, despite the underlying uncertainty. A well-known example is Idiopathic Pulmonary Fibrosis (IPF), a term that essentially describes progressive lung fibrosis of unknown origin. Nonetheless, it was already reported in 1986 that recurrent or persistent of inflammatory processes in the lung leads to IgA-mediated immune abnormalities with predominant IgA depositions (Endo and Hara, 1986). Activated neutrophils that release NETs have been implicated in the pathogenesis of fibrotic lung diseases (Zawrotniak and Rapala-Kozik, 2013). Notably, it has been demonstrated that IgA containing immune complexes are more potent to induce NETs release (Gimpel et al., 2021). The detrimental effects of excessive IgA mediated NETS release may be particularly significant in the lungs, since NETs release disrupts alveolar architecture (Mikacenic et al., 2018; King

and Dousha, 2024), causing lung injury in IPF but also in cystic fibrosis (CF) (Manzenreiter et al., 2012; Zawrotniak and Rapala-Kozik, 2013), chronic obstructive pulmonary disease (COPD) (Ladjemi et al., 2019), COVID (Barzegar-Amini et al., 2022) and post-COVID-19-fibrosis (Gomes et al., 2025) (Figure 3).

Given this, serum and mucosal IgA levels may serve as accessible biomarkers to identify a more severe subpopulation of patients with fibrotic lung disease (Ten Klooster et al., 2015; Arai et al., 2025). TGFβ is a key pathogenic cytokine in IPF (Sheppard, 2006), and has an important role in inducing IgA production (Kim and Kagnoff, 1990a) and class-switching to IgA (Kim and Kagnoff, 1990b). It was hypothesized that increased activated TGFβ may be reflected by augmented serum IgA levels. In retrospective analyses of two separate IPF cohorts, it was observed that baseline serum IgA level was a predictor for survival in IPF (Ten Klooster et al., 2015). Moreover, increased serum IgA levels were associated with more 'fibrotic active' disease and increased mortality risk. Accessing IgA in the bronchoalveolar lavage (BAL) fluid enhanced the sensitivity of the analysis and a correlation with survival was observed (Boustani et al., 2022). This has further been substantiated by the presence of IgA[+] B cells within tertiary lymphoid structures of IPF lungs (Heukels et al., 2019).

In CF, in addition to NETosis (Manzenreiter et al., 2012), an early event involves the emergence of an autoimmune response against host DNA that contributes to disease pathogenesis (Yadav et al., 2020). Notably, CF patient sera contain elevated levels of anti-dsDNA IgA autoantibodies, but not anti-dsDNA IgG. Moreover, anti-dsDNA IgA autoantibody levels in serum correlate with air-flow obstruction (Yadav et al., 2020). Anti-dsDNA IgA autoantibodies were already elevated in the blood of CF toddlers and children, and increased progressively with age, suggesting their involvement in initiation, amplification and chronic perpetuation (Yadav et al., 2020). The putative pathogenic mechanism involving IgA in CF

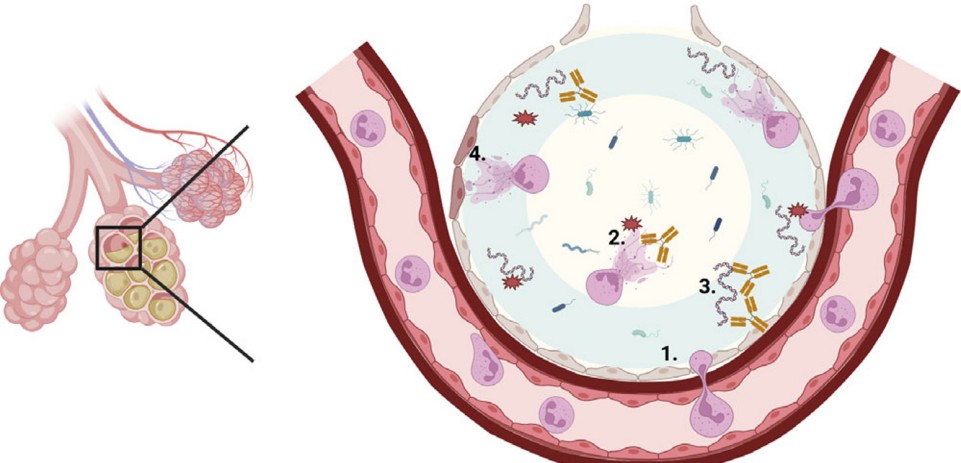

**Figure 3.** Schematic model of potential effects of IgA autoantibodies in inducing neutrophil (myeloid cell) activation and tissue damage in cystic fibrosis or idiopathic pulmonary fibrosis. Neutrophils migrate into the alveolar cavity where they get activated (1). After activation, neutrophils release NETs, which consists of DNA, citrullinated proteins, proteases and reactive oxygen species (ROS) in the mucus layer (2). NET proteases and ROS will lead to epithelial damage (4). IgA autoantibodies directed against dsDNA, citrullinated proteins or antinuclear antibodies bind to DNA and citrullinated epitopes thereby forming immune complexes (3). These IgA-antigen complexes crosslink the FcαRI on neutrophils resulting in cell activation, NETs release and release of the potent neutrophil chemoattractant leukotriene B4 (LTB4) and interleuking-8 (IL-8). This way, a neutrophil migration loop is initiated, resulting in tissue damage and fibrosis in cystic fibrosis and idiopathic pulmonary fibrosis. Picture made by Biorender.

can be distilled by integrating multiple clinical observations. Dornase alfa (Pulmozyme) is one of the most used medications to treat CF, and is a recombinant DNase administered via inhalation. It reduces mucus viscosity by degrading extracellular DNA. This breaks up sticky mucus, facilitating easy clearance of the airways of CF patients. The clinical efficacy of Pulmozyme indicates the presence of extracellular DNA in the airways of CF patients to which anti-dsDNA IgA antibodies can bind. This interaction likely forms immune complexes that induce NETs release by the abundantly present neutrophils, thereby increasing the amount of extracellular DNA and further perpetuating inflammation. Inhibition of IgA-mediated NETosis by anti-CD89 mAbs represents a rational approach to interrupt this self-amplifying mechanism and mitigate disease progression.

In COPD, accumulation of B cells and the formation of lymphoid follicles has been described. Whereas most follicular B cells were IgM$^+$ (70–80%), a two-fold increase in IgA$^+$ B cells, but not IgG$^+$, B-cell was observed in lymphoid follicles from severe COPD patients compared to control subjects (Ladjemi et al., 2019). The well-known driver of IgA class-switching, TGFβ, is elevated in COPD and described as profibrotic factor in COPD (Takizawa et al., 2001). A subset of COPD patients exhibits elevated eosinophil levels in both blood and airways, while others have varying degrees of combined neutrophil/eosinophil inflammation. Identifying patients who have both high IgA titers and high neutrophil/eosinophil numbers would assist in patient stratification for anti-CD89 mAb therapy. Furthermore, COPD and cardiovascular disease (CVD) frequently co-occur. It has estimated that between 28% and 70% of individuals with COPD also suffer from CVD (Cazzola et al., 2024). Interestingly, in middle-aged and older individuals, higher serum levels of IgA and IgG, but not IgM, are associated with CVD, cardiovascular mortality, and severe atherosclerosis (Khan et al., 2023). Inhibition of the CD89-IgA axis in this sense may therefore have added value.

Altogether, the use of (autoantigen-specific) IgA serum levels as companion diagnostic to stratify patients from various IgA-mediated pulmonary fibrotic diseases would be justified for treatment with an antagonist anti-CD89 antibody to block potential IgA mediated pathogenic effector functions.

## Liver diseases; examples of mucosal disease indications for anti-CD89 therapy

The liver, being the first organ to receive gut-derived IgAs via the portal vein, is directly exposed to circulating mucosal IgA and a target for potentially cross-reactive IgA antibodies that can bind to hepatic tissues. The role of IgA in liver diseases, especially alcoholic liver disease (ALD), non-alcoholic fatty liver disease (NAFLD) and non-alcoholic steatohepatitis (NASH) has been previously reviewed (Inamine and Schnabl, 2018). Disrupted intestinal homeostasis has been associated with liver pathologies, like liver cirrhosis (Chen et al., 2011), NASH (Zhu et al., 2013) and hepatic encephalopathy (Bajaj et al., 2012). Furthermore, a positive correlation was observed between serum IgA levels and the stage of fibrosis in NASH/NAFLD (Maleki et al., 2014; McPherson et al., 2014), suggesting a potential role for IgA in disease progression. Despite a growing body of associative evidence, mechanistic studies specifically investigating how IgA contributes to liver pathology are mostly lacking. In primary sclerosing cholangitis (PSC), which is a long-term progressive disease of the liver and gallbladder, characterized by inflammation and scarring of the bile ducts, the involvement of IgA appears more compelling. PSC is associated with intestinal disbalance. Between 3.0% to 7.5% of individuals with ulcerative colitis develop PSC, and approximately 80% of PSC patients have some form of IBD (Kummen et al., 2013), suggesting a mucosal origin of IgA. Secreted IgA, produced by plasma cells lining the biliary duct system, is the predominant immunoglobulin in bile and plays an important role in defending the biliary tract against invading intestinal pathogens (Woof and Kerr, 2006). Biliary epithelial cells are the major transporters of IgA into bile (Chen et al., 2008). In PSC, IgA autoantibodies against biliary epithelial cells correlated with disease severity (Berglin et al., 2013). Recently anti-gliadin and anti-F-actin IgA identified a subset of PSC patients with more severe disease and at risk for shortened transplantation-free survival in a large Polish cohort (Wunsch et al., 2025). Several studies reported a link between anti-Glycoprotein 2 (GP2) IgA and PSC disease severity (Jendrek et al., 2017; Wunsch et al., 2021). The presence of GP2, the antigen recognized by these antibodies, was

confirmed in the biliary tract, the site of inflammation, where it was present both in bile and in gallstones. In addition, it was shown that GP2 was synthesized in the peribiliary glands of PSC patients, supporting a pathogenic role for biliary GP2 in PSC (Lopens et al., 2024). Due to the liver's high perfusion rate, CD89-expressing peripheral neutrophils and monocytes can easily access the liver via the systemic circulation and sense tissue-bound IgA autoantibodies. Furthermore, we showed that the liver resident macrophages, i.e. Kupffer cells also express CD89 (van Egmond et al., 2000), and can be activated by IgA autoantibodies. Effector functions of CD89 bearing resident and peripheral immune cells may cause local pathogenic inflammation and damage. Interestingly, granulocyte and monocyte adsorptive apheresis (GMA), a therapy with proven efficacy in treating intestinal lesions in ulcerative colitis, has also been applied in PSC patients who have co-existing IBD. In these patients GMA was found to improve levels of alkaline phosphatase and aspartate transaminase (high levels are a poor prognosis factor). These findings suggest that GMA may serve as a treatment option in PSC, further pointing to a central role of activated myeloid lineage leukocytes from peripheral blood to disease pathology (Ito et al., 2024). The co-localization of the antigenic target GP2 at the site of inflammation along with the presence of anti-GP2 IgA antibodies that correlate with disease severity and the involvement of circulation myeloid cells, support the IgA-CD89 axis as a plausible pathogenic mechanism in PSC.

### Neurological diseases; examples of peripheral immune responses in the immune privileged central nervous system

The blood–brain barrier (BBB) protects the central nervous system (CNS) and mediates the immune-privileged status of the brain. Destabilization of the BBB occurs in several pathological conditions, including Alzheimer's disease (AD) and multiple sclerosis (MS). In these diseases, peripheral immune cells and autoantibodies can infiltrate the CNS at sites where the BBB is disrupted and lesions are located. For this reason, it is advisable to measure systemic (peripheral) IgA responses to stratify patients suffering from IgA-mediated neurological diseases (Pröbstel et al., 2020; Pocevičiūtė et al., 2022). Alternatively, although more invasive and thus more challenging, IgA levels in cerebrospinal fluid (CSF) could provide important diagnostic information, because IgA-producing cells may be located in or nearby CNS lesions (Pröbstel et al., 2020). Since only a subset of patients with these conditions exhibit high IgA autoantibodies, a companion diagnostic is crucial for selection of the appropriate patients who may benefit from blocking CD89.

AD is a neurodegenerative brain disorder characterized by beta-amyloid plaques, tau pathology, neuroinflammation, neurodegeneration, and cerebrovascular dysfunction. Chronic inflammation, BBB disruption, and increased levels of inflammatory mediators promote the penetration of immune cells into the brain, potentially contributing to neuronal death and cognitive decline in AD patients, which is a major global health concern. Microglia, the resident macrophage-like immune cells of the CNS, play a crucial role in AD (Zhang et al., 2024). In response to pathology, they upregulate MHCII, CD89, CD86, CD40, CD11a, CD54 and CD58, which are markers of antigen presentation and activation (reviewed in [Arcuri et al., 2017]). This supports that microglial cells may present antigen and are able to activate T cells (Arcuri et al., 2017). In addition, peripheral immune cells, including neutrophils, T and B lymphocytes, NK cells, and monocytes in AD (Zhang et al., 2024) can infiltrate the brain. Neutrophil accumulation has been linked to

cognitive impairment by modulating disease-relevant neuroinflammatory and vascular pathways in AD (Zenaro et al., 2015). Recent evidence indicates that vascular changes may drive neutrophil adhesion and NETs release, whereas neutrophil-derived MPO may lead to vascular oxidative stress (Smyth et al., 2022). Activated neutrophils can accumulate and adhere to vascular walls, potentially obstructing blood flow. They can also enter brain tissue through a compromised BBB and worsen disease by releasing inflammatory cytokines and NETs (Zenaro et al., 2015; Pietronigro et al., 2017). This systemic inflammatory response coupled with a compromised BBB potentially activates even more neutrophils, thereby facilitating their infiltration into the brain parenchyma (Bowman et al., 2007; van de Haar et al., 2016). Interestingly, the major chemoattractant LTB4 for neutrophils is significantly higher in AD brain, while increased ROS and NETs in circulation reflect heightened systemic neutrophil activation (Dong et al., 2018; Smyth et al., 2022; Do et al., 2023).

The contribution of autoantibodies to neurodegeneration has long been suspected with increased antibody titers linked to dementia for diverse autoantibodies, including those against GM1, adrenergic receptors, N-methyl-d-Aspartate receptor (NMDAR), tau, neurofilaments, b-amyloid (Aβ), GFAP, and neurotransmitters (Colasanti et al., 2010). Interestingly, decreased anti-Gal IgM and IgG coincided with increased anti-Gal IgA in AD patients, reflecting class switching of anti-Gal B cells from the production of "natural" IgM antibodies to an IgA-mediated adaptive response (Angiolillo et al., 2021). Moreover, IgA antibodies directed against NMDAR are detected in 10% of AD patients versus 2.8% of healthy controls (HC) (Prüss et al., 2012; Doss et al., 2014). Elevated levels of plasma IgA were mainly seen in *APOEε4* non-carriers, and were associated with cognitive decline, CRP, Aβ pathology, and brain IgA immunoreactivity. Moreover, AD patients also demonstrated higher local IgA deposit area fraction and IgA+ cell number in the hippocampus compared to healthy controls. These associations were absent in *APOEε4* carriers (Pocevičiūtė et al., 2022). This predisposition together with systemic IgA levels might provide a patient stratification strategy for interference of the IgA-CD89 axis.

Multiple sclerosis (MS) is a chronic neuroinflammatory disease of the CNS that results in progressive disability. Peripheral immune cells, particularly autoreactive T and B cells, invade the CNS at BBB breaches and initiate inflammation, myelin destruction and significant neurological disability (Haase and Linker, 2021). New approaches are needed to better understand MS pathogenesis, to have improved treatment targets and for identification of patients with poor prognosis. B cells play a central role in both MS relapses and progression, as shown by the efficacy of peripheral B cell depletion therapies and the presence of B cell aggregates in MS brains (Hauser et al., 2008). The presence of lymphoid follicle-like structures in the cerebral meninges of some MS patients supports that B-cell maturation can be sustained locally within the CNS and may contribute to the establishment of a compartmentalized humoral immune response (Hauser et al., 2008). Moreover, a key feature is the presence of oligoclonal bands in the cerebrospinal fluid in MS, indicating local antibody production by clonally restricted intrathecal B cells, although the pathogenic role of these antibodies remains to be elucidated (Link and Huang, 2006). IgG accounts for the majority of the antibodies produced in the CNS, but elevated intrathecal synthesis of both IgA and IgM has also been observed in MS patients (Sindic et al., 1984; Link and Huang, 2006; Muñoz et al., 2022). The role of intrathecal IgA production in MS is still unclear. It was shown that IgA levels in CSF predicted the rate of cortical atrophy and disease progression (Kroth et al., 2019).

**Table 2.** Disease indications and IgA's role in it

| Disease indication | IgA's role | Biomarker evidence | Therapeutic implications |
|---|---|---|---|
| Linear IgA Dermatosis | Deposits in skin leading to neutrophil attraction resulting in inflammation and blisters | Collagen type 17 IgA present in serum and in skin lesions | Decreasing/preventing neutrophil attraction to the skin and thereby inflammation and blister formation |
| IgA vasculitis | Deposits in small vessels leading to inflammation and leakage, effect observed in skin, joints, intestines and kidneys. | IgA deposits present in small vessels, IgA react against the endothelial cells. | Decreasing/preventing inflammation and leakage |
| IgA nephropathy | Accumulation in kidneys, leading to inflammation and damage of glomeruli, small vessels which are responsible for filtering waste from the blood. | IgA deposits present in glomeruli (small blood vessels) of kidneys | Decreasing/preventing IgA deposits in the kidney, resulting in less inflammation and damage to glomeruli, ultimately leading the improved kidney function. |
| Rheumatoid arthritis | Complexed auto-IgA (directed against citrullinated or carbamylated antigens, dsDNA or RF) in synovial joints, attract neutrophils into the synovium or activates osteoclasts, leading to inflammation and joint damage. | Various IgA autoantibodies present in serum and synovial fluid. | Decreasing/preventing IgA mediated inflammation and damage of e.g. the joints leading to less disability. |
| Cystic fibrosis | Complexed anti-dsDNA IgA in the lung induces the attraction of neutrophils to the lung leading to inflammation and airflow obstruction. | Anti-dsDNA IgA autoantibodies present in serum and BAL fluid | Decreasing/preventing IgA mediated inflammation and damage of the lungs, thereby improving airflow. |
| COPD | Increase in IgA$^+$ B cells in lymphoid follicles of severe COPD patients. | Increased total IgA levels in serum or BAL fluid (?) and increased neutrophil numbers in the airways. | Decreasing/preventing IgA mediated inflammation and damage of the lungs thereby improving airflow. |
| Liver diseases (e.g. primary sclerosing cholangitis) | Complexed IgA on biliary epithelial cells activate (liver resident) Kupffer cells or neutrophils and monocytes leading to liver inflammation and damage (cirrhosis) | IgA autoantibodies against biliary epithelial cells or anti-glycoprotein 2 | Decreasing/preventing IgA mediated inflammation and damage of the liver |
| Alzheimer disease | Not understood in AD, but might also play a role in attracting neutrophils into the brain. | Increased IgA serum levels and IgA autoantibodies against GM1, adrenergic receptors, N-methyl-d-Aspartate receptor (NMDAR), tau, neurofilaments, b-amyloid (Aβ), GFAP, and neurotransmitters | Decreasing/preventing IgA mediated inflammation and damage in the brain, leading to less dementia. |
| Multiple sclerosis | It remains to be determined to what extent local IgA production and the presence of specific IgA producing B cells contribute to brain pathology and disease progression | IgA autoantibodies against myelin. | Decreasing/preventing IgA mediated inflammation and damage of the central nervous system, leading to less disability |

However, recently it was observed that intrathecal synthesis of IgG and IgM but not of IgA was elevated in MS, which corresponded to an increased number of IgG$^+$ and IgM$^+$ B cells in MS meninges (Rodriguez-Mogeda et al., 2024). This latter finding is in contrast with the observation describing trafficking of IgA$^+$ plasma cells from the gut to the CNS during an MS relapse (Pröbstel et al., 2020). This was also described in mouse experimental autoimmune encephalomyelitis and hypothesized to represent a protective immune-regulatory role (Rojas et al., 2019). However, due to the absence of CD89 in mice this observation cannot be translated to human MS. Interestingly, CD89 was identified as treatment-related gene in a MS immunomodulatory treatment signature (Achiron et al., 2004). Furthermore, it was demonstrated that IgA- and IgG-positive plasma cells are abundantly present in MS lesions. IgA antibodies were localized on axons in MS plaques and may contribute to axonal damage (Zhang et al., 2005). Additionally, MS patients with IgA autoantibodies against myelin basic protein (Schumacher et al., 2019) or myelin oligodendrocyte glycoprotein (Ayroza Galvão Ribeiro Gomes et al., 2023) have been described. Nonetheless, it remains to be determined to what extent local IgA production and the presence of specific IgA producing B cells contribute to brain pathology and disease progression in MS. However, the presence of IgA-producing cells and the presence of CD89 expressing cells like monocytes, macrophages, neutrophils and microglia in MS lesions, is

suggestive for a role of CD89-IgA axis in MS. An overview of all diseases discussed in this review is summarized in Table 2.

## Conclusion

It is striking that the link between high serum levels of disease-specific IgA and disease severity was already established over 30 years ago, yet this knowledge was not translated for clinical practice. Although autoimmune and fibrotic disorders are heterogenous by nature, and patients may have multiple immune pathways contributing to pathology, the yet untargeted IgA-CD89 axis may contribute variably depending on the patient's disease status. The involvement of this axis in multiple autoimmune or fibrotic diseases may justify the use of phase 2a clinical backet-studies, similar to those already employed in the development of oncology therapeutics. Despite clear evidence that IgA is a very potent activator of inflammation and driver of severe disease, it has not been exploited either as therapeutic target or as biomarker for patient stratification. By developing a therapeutic antibody against CD89, disease specific IgA characteristics can be used to select patients in various diseases eligible for anti-CD89 therapy, resulting in a precision medicine guided patient-centric approach, as only patients with IgA autoantibodies and IgA mediated disease will be sensitive to treatment with a CD89 antagonist.

In this review, we propose a novel approach for the development of new drugs for inflammatory diseases: starting with the identification of an accessible, relevant biomarker in treatment refractory patients, namely IgA. Only after the biomarker has been identified, the therapeutic target and strategy should be selected in order to ensure the most optimal approach.

**Open peer review.** To view the open peer review materials for this article, please visit http://doi.org/10.1017/pcm.2025.10005.

**Acknowledgements.** We thank Jurre Jansma for his help with preparing the table "Approved (EU/US) and phase 3 therapies" and preparing the graphical abstract.

**Author contribution.** MvD and LB wrote the first draft of the manuscript. AC and MvE provided feedback on the draft. MvD finalized the manuscript.

**Financial support.** Since MvD, AC and LB are directly or indirectly employed by JJP Biologics they obtained financial support for writing the manuscript.

**Competing interests.** LB is board member of JJP Biologics which develops an anti-CD89 antibody.

AC is an employee of JJP Biologics which develops an anti-CD89 antibody.

LB and MvD are employed by Leidenlabs BV which performs experiments as sub-contractor for JJP Biologics which develops an anti-CD89 antibody.

MvE is a scientific advisor of JJP Biologics which develops an anti-CD89 antibody.

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
