## [Reviewer Report]

<b>Abstract, 30-32</b>

*“Here, we present an alternative approach to facilitate and secure biomarker development before clinical development is initiated, enabling patient stratification in early clinical studies.”*

Alternative to what? Many cancer drugs do in fact have a defined biomarker and select patients based on the biomarker right from early phase trials. E.g. drugs targeting KRAS mutation- cancer somatic mutations can be detected in the blood using CDx assay.

The authors might be coming at this from an indication specific context, so it might be helpful to narrow down the scope from the beginning. I fully agree that in non-cancer diseases, biomarker first approaches are very uncommon, and it is likely the intention of this manuscript to call attention to IgA-mediated inflammatory diseases. It would give the paper much credit to set the stage by acknowledging and contrast the different degrees to which precision/personalized treatment has progressed for different disease types, and then call attention to their diseases of interest.

The words “facilitate and secure” also made me think that the paper was going to present pre-clinical or clinical trial data from their work, but the bulk of the manuscript is a review of IgA mediated diseases. Suggest to re-word this part.

I understand that the authors are trying to justify their concept of starting out on the drug development path by first understanding their biomarker of interest. This should be embedded in the title and abstract.

<b>p.3, 4-8 </b>

The use of the GPS metaphor seems a bit far fetched and lengthy for a scientitic paper. If this metaphor resurfaces throughout the review, it could be warranted, but this is not the case. Suggest to shorten or remove.

<b>p.3, 21-22 </b>

*“Moreover, the use of companion diagnostics to facilitate a patient-centric approach remains disappointing (3). This is mainly caused by the timing…”*

It is not clear what context the authors are referring to by “Patient centric approach” - Is this referring to in the real-world clinic setting? In drug development?

Suggest to narrow or specify your scope to “in the context of drug development”.

lag time and/or failure to utilize CDx in clinical practice setting is multifactorial, many regulatory and real-world /practical challenges also exist.

<b>P3, 29-31</b>

Again the paper is a lit review of disease indications and does not really describe how biomarker development was “facilitated and secured”. For example, is there going to be efforts to develop auto-antigen specific IgA assays for the indications that do not alread have a test? Have authors characterized and validated the threshold / cutoff for auto-IgA titre?

Will the clinical trials then aim to select patients based on their auto-antibody titre?

If IgA titre is the clinical “biomarker”, then based on FDA definition, one would expect the analytical and clinical utility of this to be assessed and validated. Therefore I am not comforatble with the phrase “facilitate and secure biomarker development before clinical development is initiated”. The content of this manuscript does not cover this adequately.

<b>

Overall comment about disease animal models –</b>

It is not clear from language if these were a review of literature or if this was talking about authors own work. From a look at some of the the references it seems like this is describing their own work, so this should be made more clear by using pronouns like “we”.

<b>Overall comment about the review of each disease type</b>

Would be highly valuable to the reader if authors could summarize what the drug development landscape for these conditions look like – perhaps in a table – what drugs are approved, what is under clinical development, what pathways these drugs target, then highlight that IgA-CD89 axis as a potential novel addition to the space.

The current textual content is a bit lengthy for me and takes me away from the main point of the review.

<b>

Conclusion, p15, 57-58</b>

Would give the paper merit to summarize the current landscape of therapeutic drugs targeting IgA-mediated inflammatory conditions. Seems to be some drugs out there for IgAN. I feel that authors should be aware that immune diseases can be very heterogenous, and exercise caution in their wording.

Statements should preferably be made with an underlying appreciation of the (most likely) fact that targeting 1 single pathway will most likely NOT treat or restore equlibrium to diseases of complex dysregulation, for all patients (precisely and further highlighting the concept of inter-individual differences).

<b>P16, 4-5</b>

Do authors consider writing about a basket trial to concurrently evaluate their candidate in multiple disease conditions?

<b>P16, 8-10</b>

This is repetitive, definition of precision medicine should already be clear in the intro.

<b>P16, 13-15</b>

Can the authors reveal more about how they are validating their biomarker of interest target in the many indications they have reviewed?

---

## [Reviewer Report]

This review has a relevant concept, however, there are several comments that should be addressed to improve its scientific rigor:

Major points:

Please include a short paragraph describing your literature retrieval strategy. For guidance on structuring this section, you may use Gasparyan AY, et al. (2011). Writing a narrative biomedical review: considerations for authors, peer reviewers, and editors. Rheumatology International. 2011;31(11):1409–1417. doi:10.1007/s00296-011-1999-3.

Please consider adding one summary table comparing IgA’s role, biomarker evidence, and therapeutic implications across diseases.

The reference list requires an update.

Please include the COI statement and Author Contribution in line with CrediT.

Minor points:

Please carefully recheck reference formatting (some citations are missing years or having inconsistent styles)

---

## [Editor Report]

The reviewers have seen merit in your work and suggested some changes that are relevant and will improve the manuscript. The suggested changes should be straightforward to address.

---

## [Reviewer Report]

I thank the authors for considering my feedback and applaud their effort in summarizing the literature. I loved the addition of the tables as it provides a birds eye view of their targeted landscape. I support their concept and hope this article will contribute to advocating for precision medicine.

---

## [Reviewer Report]

The reference list still contains several outdated sources that should be replaced with more recent publications.

---

## [Editor Report]

Thank you for submitting a revised manuscript. The comments of the reviewers have been addressed. 

One additional comment from me relating to the introduction: ERBB3 (the correct symbol for HER3) expression is not used to select pts for cetuximab therapy; please change text to: “...while tumor expression of (wild type) human epidermal growth factor receptor 2 (ERBB2) is used to select patients for trastuzumab and other ERBB2-directed therapies.” Along with the KRAS example, this will suffice to illustrate use of biomarkers in cancer.